# Probiotics and *Achyranthes bidentata* Polysaccharides Improve Growth Performance via Promoting Intestinal Nutrient Utilization and Enhancing Immune Function of Weaned Pigs

**DOI:** 10.3390/ani11092617

**Published:** 2021-09-07

**Authors:** Gaifeng Hou, Wei Peng, Liangkai Wei, Rui Li, Xingguo Huang, Yulong Yin

**Affiliations:** 1Key Laboratory of Agro-Ecological Processes in Subtropical Region, Hunan Provincial Key Laboratory of Animal Nutritional Physiology and Metabolic Process, Hunan Research Center of Livestock and Poultry Sciences, South Central Experimental Station of Animal Nutrition and Feed Science in the Ministry of Agriculture, National Engineering Laboratory for Poultry Breeding Pollution Control and Resource Technology, Institute of Subtropical Agriculture, Chinese Academy of Sciences, Changsha 410125, China; hougf521@163.com (G.H.); yinyulong@isa.ac.cn (Y.Y.); 2College of Animal Science and Technology, Hunan Agricultural University, Changsha 410128, China; pengwei20210730@163.com (W.P.); wlk20210730@163.com (L.W.); hxg68989@hunau.edu.cn (X.H.)

**Keywords:** probiotics, *Achyranthes bidentata* polysaccharides, growth performance, immune, weaned pigs

## Abstract

**Simple Summary:**

It is frequent to see that in-feed antibiotics are added to piglets diets because of their enteric problems after weaning. However, in-feed antibiotics have been forbidden to used in livestock production since 1 July 2020 in China. Therefore, it is urgent to develop some promising alternatives to in-feed antibiotics. Probiotics and plant extracts are considered to be the potential replacements, which have been studied or applied in animal production. In our study, we found that probiotic or *Achyranthes bidentata* polysaccharides used alone or in combination, the combination augmenting the positive effect more than the independent supplement, could improve piglets’ growth performance via promoting intestinal nutrient digestion and absorption and enhancing immune funtion, and the beneficial role was comparable to that of the selected in-feed antibiotics.

**Abstract:**

The experiment aimed to investigate the effects of probiotics and *Achyranthes bidentata* polysaccharides on the growth performance, nutrients digestibility, and immune function of weaned pigs. One hundred and twenty weaned pigs (about 7 kg BW, 23 ± 2 d) were allotted to five dietary treatments (CON: antibiotics-free basal diet; ANT: CON + antibiotics; PRO: CON + probiotics; ABPS: CON + *Achyranthes bidentata* polysaccharides; P-ABPS: PRO + ABPS) for a 28-day trial. Compared with CON, pigs in ANT, PRO, ABPS, and P-ABPS had greater (*p* < 0.05) ADG, ATTD of CP and GE, serum ALB, IgA and IL-2, duodenal intraepithelial lymphocyte, ileal VH and jejunal mucosa sIgA, but lower (*p* < 0.05) fecal scores, serum BUN, and IL-1β. Meanwhile, ANT, PRO, ABPS, and P-ABPS exhibited similar beneficial roles on growth performance, nutrients digestibility, serum parameters, and immune function. Interestingly, P-ABPS effects were similar to those obtained with ANT rather than with PRO or ABPS. In conclusion, Dietary PRO or ABPS used alone or in combination (P-ABPS), the combination augmenting the positive effect more than the independent supplement, could improve piglets’ growth performance via promoting intestinal nutrient digestion and absorption and enhancing immune function, indicating it had the potential to act as an alternative to in-feed antibiotics used in piglet diets.

## 1. Introduction

Early weaning can increase the pigs weaned per sow per year (PSY) and improve the utilization efficiency of pigsties, which has been commonly applied in the modern swine industry all over the world. However, early weaning might result in piglet anorexia, growth retardation, severe enteric infections, or even death, due to shorter suckling period as well as abrupt changes in dietary and social environment in comparison with the natural weaning [1,2]. Traditionally, adding in-feed antibiotics to the piglet diets can effectively control intestinal diseases and promote growth. However, the ban on the use of in-feed antibiotics in livestock feed has been proposed and gradually put into practice worldwide because of their side effects on human health and the environment [3]. Therefore, finding safe and efficient antibiotic alternatives for young pigs has become a global focus.

Probiotics known as an in-feed antibiotics replacement have been widely used in piglet diets, which exert beneficial effects on regulation of intestinal microflora, control of enteric pathogenic bacteria, ameliorating gut health, reducing diarrhea, and improving growth performance [4,5]. Additionally, traditional Chinese herbal medicine is also a potential substitute to antibiotics. Previous studies have confirmed that polysaccharides derived from the Chinese herb ox knee *Achyranthes bidentata* (ABPS) possesses immuno-modulatory functions, and can be used as a diet additive for weanling piglets to enhance piglet growth [6,7].

Based on the above, we hypothesized that probiotics in combination with ABPS would improve piglets’ growth and its beneficial effect would be better than when used alone, and equal or superior to in-feed antibiotics. To test the hypothesis, growth performance, nutrients digestibility, serum biochemical and immune indexes, serum cytokines, intestinal morphology, and immune function of piglets were determined.

## 2. Materials and Methods

All animal care, handling, and surgical techniques followed protocols approved by the Animal Care and Use Committee of the Institute of Subtropical Agriculture, Chinese Academy of Sciences (IACUC#201302).

The chlortetracycline hydrochloride belongs to antibiotics growth promoters (AGP), which is provided by the Chia Tai Group (Changsha, China) and its available content is 15%. The probiotics used in this study were made by our lab and the product has been patented in China [8]. The product mainly included *Lactobacillus delbrueckii*, *Bacillus subtilis* and *Saccharomyces paradoxus*, and the total colony-forming units (CFU) were 1.0 × 10^8^ CFU/g. *Achyranthes bidentata* Polysaccharides (ABPS) were purchased from Wuhan Dongkang Source Technology Co., Ltd. (Wuhan, China), the products are prepared by an extraction and drying process, and its purity (99%) was measured by HPLC.

### 2.1. Animals, Experimental Design, Diets and Management

A total of 120 crossbred (Landrace × Large White), weaned pigs with an initial body weight of about 7 kg (23 ± 2 d) were randomly assigned to 1 of 5 diets: (1) non-antibiotic basal diet (CON); (2) basal diets + 75 mg/kg chlortetracycline hydrochloride (ANT); (3) basal diets + 1000 mg/kg probiotics (PRO); (4) basal diets + 500 mg/kg *Achyranthes bidentata* polysaccharides (ABSP); and (5) basal diets + 1000 mg/kg probiotics + 500 mg/kg *Achyranthes bidentata* polysaccharides (P-ABPS). Each dietary group had 6 pens (1.4 m × 2.0 m), and 4 pigs (the same sex ratio) were included in each pen. All piglets were housed in a temperature-controlled (28 ± 2 °C) nursery room. The experimental period was 28 days, and pigs had free access to feed and water in the whole period. The piglets were fed their respective diets 4 times per day at 8:00 a.m., 11:00 a.m., 2:30 p.m. and 5:30 p.m., successively. The basal diets (Table 1) were formulated to meet or exceed the nutrient requirements for weaned pigs recommended by the NRC [9] and also comply with the practice of Chinese swine production.

### 2.2. Growth Performance and Fecal Score

Pig’s body weights were measured at the beginning and end of the experimental period, and feed consumption was recorded on a pen basis during the experiment to calculate ADG, ADFI, and F/G. Fecal score was performed according to the method introduced by Lu et al. [10], and scoring criteria was showed as the following: hard feces = 1; slightly soft = 2; soft and partially formed feces = 3; loose, semi-liquid = 4; watery feces = 5.

### 2.3. Nutrient Digestibility

During a 3-day (d 25–27) collection period, fecal samples from each pen were partially collected and transferred into plastic bags and stored at −20 °C. Three-day fecal samples were separately thawed and pooled within pen and diet, homogenized and subsampled. Fecal subsamples were dried at 65 °C for 72 h in an air-forced oven. The diets and fecal samples were finely ground to pass through a 1-mm screen and analyzed using the AOAC [11] procedures for dry matter (DM, 930.15) and crude protein (CP, 984.13). The gross energy (GE) of samples were determined with an Oxygen Bomb Automatic Calorimeter (HXR-6000, Hunan Huaxing Energy Sources Instrument Co. Ltd., Changsha, China). The acid-insoluble ash (AIA) [12] was used as an internal indicator to calculate apparent total tract digestibility (ATTD) of DM, CP, and GE in diets fed to pigs.

### 2.4. Serum Biochemical and Immune Indexes

At the conclusion of the trial (d 29), one pig from each pen with similar BW was selected to collect blood from the anterior vena cava using 10 mL vacuum tubes. Blood samples (5 mL) were left alone for 30 min, then centrifuged at 3000 r/min and 4 °C for 10 min to collect the serum, and stored at −20 °C for subsequent analysis. Serum total protein (TP), albumin (ALB), blood urea nitrogen (BUN), immune globulin A (IgA), immune globulin G (IgG), and immune globulin M (IgM) were analyzed using BS 200 Automatic Biochemical Analyzer (Mindray) with the relevant reagent kits and its instructions (Mindray).

### 2.5. Serum and Intestinal Cytokines Mesured by Elisa Kit

Serum interleukin-1β (IL-1β), interleukin 2 (IL-2) and Tumor necrosis factor-α (TNF-α) were determined by radio-immunoassays using kits purchased from Beijing Chemclin Biotech (Beijing, China) as described by Kong et al. [13]. 

After blood collection, pigs were humanely slaughtered by an intracardial injection of sodium pentobarbital at a dose of 50 mg/kg BW, then the intestinal tissues were isolated. The jejunal tissue was homogenated, centrifuged, and the supernatant was achieved to collect total protein. The protein concentrations were detected using a BCA protein assay kit (MERCK). The level of jejunal secretory IgA (sIgA) was determined in triplicate by an Elisa kit (Cusabio Biotech Co., Ltd., Wuhan, China) as described by Li et al. [14]. The unit was expressed as μg/mg protein.

### 2.6. Intestinal Morphology and Intraepithelial Lymphocyte

After intestinal tissue separation, about 2 cm segments of duodenum, jejunum, and ileum were quickly snipped and flushed with sterile saline, fixed immediately in 4% paraformaldehyde solution and then embedded in paraffin. An approximate 5 μm thickness section of each sample was stained with hematoxylin-eosin (HE) for intestinal morphology analysis. The count of the intestinal intraepithelial lymphocytes was measured using HistostainTM-Plus Kits (Beijing Zhongshan Golden Bridge Company, Beijing, China). For each intestinal tissue section, the number of intraepithelial lymphocytes together with the villus height (VH) and crypt depth (CD) were calculated using computer-assisted microscopy (DT2000, Panasonic, Osaka, Japan) and image-analysis software (Motic Images Plus 2.0, Dongguan, China).

### 2.7. Statistical Analysis

All statistical analyses were performed by one-way ANOVA using the GLM procedure of SPSS 17.0 (SPSS Inc., Chicago, IL, USA). Each pen acts as a statistical unit for growth performance and nutrients digestibility, whereas each pig as a statistical unit for serum chemical parameters, intestinal structure, and immune indexes. All results present as Mean ± SEM. Difference among means were determined using Duncan’s multiple range test. *p* < 0.05 was considered statistically significant, while 0.05 < *p* < 0.10 as a tendency.

## 3. Results

### 3.1. Growth Performance and Fecal Score

Pigs in ANT, PRO, and P-ABPS groups had greater (*p* < 0.05) final weight, ADG, and ADFI than those in the CON group (Table 2). Dietary supplementation with ANT and P-ABPS decreased (*p* < 0.05) F/G and fecal score compared with CON. However, no differences (*p* > 0.05) were observed in these response criteria among ANT, PRO, and P-ABPS groups.

### 3.2. Nutrient Digestibility

Compared with CON, ANT, PRO, ABPS, and P-ABPS improved (*p* < 0.05) apparent total tract digestibility (ATTD) of CP and GE in diets fed to weaned pigs (Table 3). However, there were no differences (*p* > 0.05) among ANT, PRO, ABPS, and P-ABPS treatments.

### 3.3. Serum Biochemical and Immune Parameters and Cytokines

Serum TP in the P-ABPS group was higher than in the CON group (*p* < 0.05), but no differences were found compared with the remaining groups (Table 4). Pigs in the ANT, PRO, ABPS, and P-ABPS groups had greater (*p* < 0.05) serum albumin contents, but lower (*p* < 0.05) serum blood urea nitrogen (BUN) and interleukin-1β (IL-1β) contents than the CON group. Serum immune globulin A (IgA) and interleukin-2 (IL-2) concentrations were higher (*p* < 0.05) in pigs fed the ANT, ABPS, and P-ABPS supplemented diet in comparison with the CON diet.

### 3.4. Intestinal Morphology and Immune Function

Compared with the CON and ANT groups, jejunal villus height (VH) in ABPS and P-ABPS was improved (*p* < 0.05), and ileal VH in PRO, ABPS, and P-ABPS groups had a trend to increase (*p* = 0.081) (Table 5). Intraepithelial lymphocyte in duodenum was greater (*p* < 0.05) in ANT, PRO, ABPS, and P-ABPS group than in CON group. Dietary P-ABPS improved (*p* < 0.05) intraepithelial lymphocyte in jejunum and ileum in comparison with the CON diet. Additionally, sIgA in the jejunal mucosa of pigs fed ANT, ABPS, and P-ABPS diets was greater (*p* < 0.05) than the CON diets (Figure 1).

## 4. Discussion

Piglets after weaning exhibit impaired or retarded growth performance and their gastrointestinal tract are susceptible to pathogens, which may easily cause post-weaning diarrhea [10,15]. In-feed antibiotics’ supplementation can effectively solve these difficulties. In our study, compared with the CON group, pigs in the ANT group had greater ADG and ADFI, and lower F/G and fecal score; meanwhile, dietary PRO, ABPS, or P-ABPS supplementation showed a similar beneficial role on piglet growth performance as ANT. Probiotics are considered to be an alternative to in-feed antibioics and can promote piglet growth through favoring nutrient digestion, modulating the gastrointestinal ecosystem, stimulating the immune system, and protecting the host from gastrointestinal tract diseases [16,17,18]. However, some studies showed that there was no significant role for piglet growth performance in response to probiotics supplementation [19], which might be attributed to the viable counts of probiotics ingested in feed and colonized in the intestine. Just as it is defined by the World Health organization (WHO) as “microorganisms, which administering live and adequate amounts, confers a benefit to the health of the host” [20]. ABPS as a traditional Chinese medicine has also been used in pig production, and previous studies both in vitro and in vivo revealed that ABPS improved weaned pig growth performance mainly through enhancing cellular and humoral immune responses to perform an important protective role in the non-specific defense against infections [6,7]. Interestingly, probiotics in combination with ABPS used in the present study showed a better beneficial effect on the piglet growth than probiotics or ABPS used alone, which was line with our expectations and suggest the existence of an interaction of the combined use of the two products. 

Feed is the primary source of energy and protein for animal maintenance and growth requirements [21,22]. The digested and absorbed nutrients by the gastrointestinal tract provide energy and amino acids for animal life activities and growth via biological oxidation and metabolism [23,24]. In our study, dietary ANT, PRO, ABPS, and P-ABPS supplementation increased ATTD of GE and CP in diets fed to piglets, demonstrating that these products promoted gastrointestinal digestion and absorption, enhancing the nutrients’ utilization efficiency, which is an evidence for improving piglet growth performance. The improvement of nutrient utilization was chiefly associated with reducing nutrient consumption of intestinal bacteria by in-feed antibiotics [25], producing organic acids and digestive enzymes by probiotics [17], and improving intestinal immune function by ABPS [6,7].

The fluctuation of serum biochemical indexes can reflect the body metabolism and health. The levels of serum TP, ALB, and BUN are strongly linked to protein digestion and absorption, especially serum BUN, which is the main end product of protein metabolism in mammals [26]. In our study, dietary ANT, PRO, ABPS, and P-ABPS addition increased the content of serum ALB but decreased the serum BUN content, indicating that protein utilization in these dietary treatments was improved, which echoed the improved ATTD of CP and ADG aforementioned. IgA, IgG, and IgM are generated by bone marrow-derived B lymphocytes, which can mediate humoral immunity and serve as important response criteria to evaluate humoral immune response in humans and animals [27]. In the present study, pigs in the ANT, ABPS, and P-ABPS groups had greater serum IgA content than those in the CON group, indicating that pigs’ humoral immunity were boosted in response to these products’ supplementation, which concurred with the previous reports [5,6,7,28]. Furthermore, the increased protein utilization by adding these products could provide substrates for immunoglobulin synthesis, which could partly offer interpretation for the improvement of immunity.

The gastrointestinal tract is the place for digestion and absorption of nutrients, maintaining its morphological structure is of great importance for gut health. However, anorexia or low feed intake after weaning appears, which means lack of supply of luminal nutrition, and challenges and stressors associated with weaning also cause changes to the structure and function of the gastrointestinal tract [28]. In our study, pigs in the CON group had lower ADFI and villus height in the jejunum and ileum compared with pigs in P-ABPS groups, indicating that a dietary P-ABPS played a role in maintaining intestinal structure and function, which might be related to regulation of microflora by probiotics and improvement of mucosal structure and function by ABPS [6,7,28]. The weaning transition not only results in marked structural and functional changes to the small intestine, but also contributes to an intestinal inflammatory status that in turn compromises villouscrypt architecture, gastrointestinal tract barrier function, and disruption of the microbiota [10,28]. IL-1β and TNF-α, proinflammatory cytokine, are produced by mononuclear macrophages, which can reflect the body inflammatory condition. IL-2 is a chemotactic factor and mainly generated by activated T cells, which can promote growth, proliferation, and differentiation of lymphocytes to participate in humoral immune response [29,30]. In our study, pigs in the CON group had greater serum IL-1β and lower IL-2 than pigs in the other groups, suggesting that adding ANT, PRO, ABPS, and P-ABPS to the diet could alleviate inflammatory response induced by weaning.

The gastrointestinal tract is the largest immune organ and its superficial mucosa serve as the first line of defense against pathogen invasion. sIgA is the main antibody present on mucosal surfaces, which have long been considered as a first line of defense via providing passive immunoprotection against invading antigens and can protect the intestinal epithelium from enteric pathogens and toxins [14,31]. Meanwhile, intraepithelial lymphocytes (IEL) reside within the epithelial layer of mucosal and barrier tissues and represent the first immune system cells to encounter pathogens that have invaded an epithelial surface [32]. In the present study, dietary P-ABPS supplementation increased the levels of sIgA in the jejunal mucosa and intraepithelial lymphocytes in the small intestine, indicating that dietary PRO combined with ABPS could enhance intestinal immune function.

## 5. Conclusions

Dietary PRO or ABPS used alone or in combination (P-ABPS) could improve piglets’ growth performance via promoting intestinal nutrient digestion and absorption and enhancing immune function, indicating that PRO, ABPS, or P-ABPS have the potential to replace in-feed antibiotics in piglet diets, and the combined effect of probiotics and ABPS is comparable to in-feed antibiotics.

## Figures and Tables

**Figure 1 animals-11-02617-f001:**
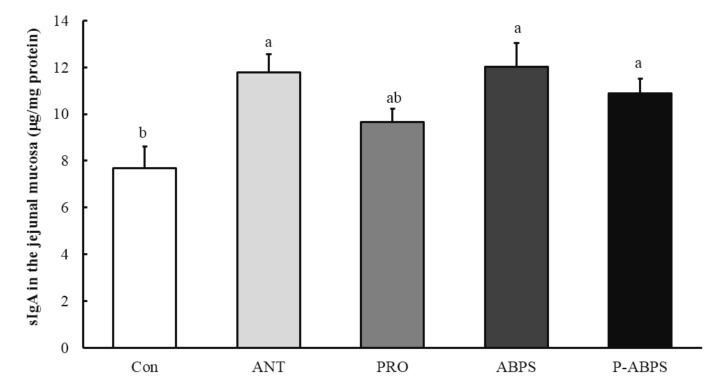
Levels of secreted immunoglobulin A (sIgA) in the jejunal mucosa. CON, basal diets; ANT, basal diets + antibiotics; PRO, basal diets + probiotics; ABSP, basal diets + *Achyranthes bidentata* polysaccharides; P-ABPS: basal diets + probiotics + *Achyranthes bidentata* polysaccharides. Different letters (a,b) above the error bars indicated significant differences (*p* < 0.05). *n* = 6 means data come from six pigs per dietary treatment.

**Table 1 animals-11-02617-t001:** The ingredient composition and nutrient levels of the basal diets (as-fed basis, %).

Items	Contents
Ingredients	
Corn	49.17
Extruded corn	16.00
Soybean meal	18.00
Fermented soybean meal	4.00
Whey powder	2.00
Glucose	2.00
Fatty powder	1.00
Fish meal	4.00
Premix ^1^	3.00
l-lysine HCl	0.50
DL-methionine	0.11
l-threonine	0.20
l-tryptophan	0.02
Total	100.00
Calculated nutrient level	
Digestible energy, kcal/kg	3546
Metabolizable energy, kcal/kg	3352
Crude protein	18.61
SID lysine	1.23
SID methionine	0.36
SID threonine	0.73
SID tryptophan	0.20
Analysed nutrient level	
Gross energy, kcal/kg	3981
Dry matter	88.86
Crude protein	18.78

Note: ^1^ Provided per kg of diet: 12,000 IU vitamin A, 2400 IU vitamin D3, 45 IU vitamin E, 3.0 mg vitamin K, 0.40 mg vitamin B1, 6.4 mg vitamin B2, 0.3 mg vitamin B6, 36 μg VB12, 2 mg folic acid, 40 mg nicotinic acid, 20 mg D-pantothenic acid, 0.45 mg biotin, 120 mg Fe, 6 mg Cu, 40 mg Mn, 100 mg Zn, 1.30 mg I, 0.30 mg Se.

**Table 2 animals-11-02617-t002:** Effects of probiotics *and achyranthes* bidentata polysaccharides on the growth performance of piglets (23–51 d).

Items	CON ^1^	ANT ^1^	PRO ^1^	ABPS ^1^	P-ABPS ^1^	SEM	*p*-Value
Initial Weight (kg)	6.89	6.84	6.83	6.88	6.85	0.47	0.872
Final Weight (kg)	13.71 ^b^	15.91 ^a^	14.53 ^a^	14.13 ^ab^	15.47 ^a^	2.31	0.045
ADG (g/d) ^2^	234.39 ^c^	323.72 ^a^	275.42 ^ab^	259.12 ^b^	308.20 ^a^	64.93	0.006
ADFI (g/d) ^2^	418.27 ^b^	476.28 ^a^	451.39 ^ab^	429.94 ^b^	489.71 ^a^	87.94	0.048
F/G ^2^	1.78 ^a^	1.47 ^b^	1.64 ^ab^	1.66 ^ab^	1.59 ^b^	0.14	0.031
Fecal score ^2^	3.47 ^a^	2.78 ^b^	2.84 ^b^	2.82 ^b^	2.65 ^b^	0.57	0.025

Note: ^1^ CON, basal diets; ANT, basal diets + antibiotics; PRO, basal diets + probiotics; ABSP, basal diets + *Achyranthes bidentata* polysaccharides; P-ABPS: basal diets + probiotics + *Achyranthes bidentata* polysaccharides. ^2^ ADG, average daily weight; ADFI, average daily feed intake; F/G, feed/gain; Fecal score: hard feces = 1; slightly soft = 2; soft and partially formed feces = 3; loose, semi-liquid = 4; watery feces = 5. ^a,b,c^ Mean values with different superscripts in the same row differ significantly (*p* < 0.05). SEM is standard error of mean. *n* = 6 means data come from six pens per dietary treatment.

**Table 3 animals-11-02617-t003:** Effects of probiotics and *Achyranthes bidentata* polysaccharides on apparent total tract digestibility (ATTD) of diet fed to piglets.

Items	CON ^1^	ANT ^1^	PRO ^1^	ABPS ^1^	P-ABPS ^1^	SEM	*p*-Value
Dry matter	88.27	89.33	88.82	88.73	89.11	1.24	0.231
Crude protein	78.40 ^b^	82.44 ^a^	82.36 ^a^	82.27 ^a^	82.39 ^a^	2.31	0.032
Gross energy	85.52 ^b^	87.54 ^a^	87.36 ^a^	87.27 ^a^	87.27 ^a^	0.49	0.027

Note: ^1^ CON, basal diets; ANT, basal diets + antibiotics; PRO, basal diets + probiotics; ABSP, basal diets + *Achyranthes bidentata* polysaccharides; P-ABPS: basal diets + probiotics + *Achyranthes bidentata* polysaccharides. ^a,b^ Mean values with different superscripts in the same row differ significantly (*p* < 0.05). SEM is standard error of mean. *n* = 6 means data come from six pens per dietary treatment.

**Table 4 animals-11-02617-t004:** Effects of probiotics and *Achyranthes bidentata* polysaccharides on serum biochemical and immune indexes, and cytokines of piglets.

Items	CON ^1^	ANT ^1^	PRO ^1^	ABPS ^1^	P-ABPS ^1^	SEM	*p*-Value
Total protein (g/L)	46.75 ^b^	51.10 ^ab^	52.70 ^ab^	55.15 ^ab^	59.50 ^a^	5.30	0.037
Albumin (g/L)	23.98 ^b^	31.09 ^a^	29.51 ^a^	29.77 ^a^	28.39 ^a^	2.17	0.019
Blood urea nitrogen (mmol/L)	4.60 ^a^	3.74 ^b^	3.53 ^b^	3.62 ^b^	3.71 ^b^	0.84	0.031
Immune globulin A(g/L)	0.84 ^b^	0.97 ^a^	0.88 ^ab^	1.05 ^a^	1.12 ^a^	0.22	0.016
Immune globulin M(g/L)	0.53	0.84	0.70	0.76	0.87	0.16	0.238
Immune globulin G(g/L)	0.98	1.08	1.12	1.16	1.2	0.20	0.192
Interleukin-2(ng/mL)	2.37 ^b^	4.97 ^a^	3.45 ^ab^	3.79 ^a^	5.35 ^a^	0.94	0.049
Interleukin-1β(ng/mL)	0.27 ^a^	0.21 ^b^	0.19 ^b^	0.17 ^b^	0.18 ^b^	0.03	0.031
Tumor necrosis factor-α(ng/mL)	0.51	0.49	0.52	0.46	0.47	0.07	0.107

Note: ^1^ CON, basal diets; ANT, basal diets + antibiotics; PRO, basal diets + probiotics; ABSP, basal diets + *Achyranthes bidentata* polysaccharides; P-ABPS: basal diets + probiotics + *Achyranthes bidentata* polysaccharides. ^a,b^ Mean values with different superscripts in the same row differ significantly (*p* < 0.05). SEM is standard error of mean. *n* = 6 means data come from six pigs per dietary treatment.

**Table 5 animals-11-02617-t005:** Effects of probiotics and *Achyranthes bidentata* polysaccharides on small intestinal morphology and intraepithelial lymphocyte in piglets.

Items	CON ^1^	ANT ^1^	PRO ^1^	ABPS ^1^	P-ABPS ^1^	SEM	*p*-Value
Duodenum							
Villus height	412.37	408.42	415.17	423.29	419.34	8.81	0.221
Crypt depth	173.07	169.11	171.45	181.22	177.09	4.80	0.429
VH/CD	2.38	2.42	2.42	2.34	2.37	0.04	0.141
Intraepithelial lymphocyte	39.67 ^c^	83.00 ^b^	93.67 ^a^	90.00 ^a^	95.33 ^a^	17.85	0.009
Jejunum							
Villus height	387.21 ^b^	385.07 ^b^	390.19 ^ab^	397.92 ^a^	394.83 ^a^	5.31	0.034
Crypt depth	139.41	137.34	141.15	146.96	144.13	3.81	0.262
VH/CD	2.78	2.80	2.76	2.71	2.74	0.04	0.901
Intraepithelial lymphocyte	48.75 ^b^	79.30 ^ab^	88.30 ^ab^	81.70 ^ab^	94.64 ^a^	19.44	0.013
Ileum							
Villus height	322.52 ^b^	328.33 ^b^	345.00 ^a^	337.74 ^a^	341.51 ^a^	9.36	0.081
Crypt depth	120.54	119.27	126.38	120.09	128.10	4.06	0.462
VH/CD	2.68	2.75	2.73	2.81	2.67	0.06	0.561
Intraepithelial lymphocyte	67.27 ^b^	86.41 ^ab^	91.32 ^ab^	89.20 ^ab^	94.16 ^a^	9.12	0.032

Note: ^1^ CON, basal diets; ANT, basal diets + antibiotics; PRO, basal diets + probiotics; ABSP, basal diets + *Achyranthes bidentata* polysaccharides; P-ABPS: basal diets + probiotics + *Achyranthes bidentata* polysaccharides. ^a,b,c^ Mean values with different superscripts in the same row differ significantly (*p* < 0.05). SEM is standard error of mean. *n* = 6 means data come from six pigs per dietary treatment.

## Data Availability

All data are included in the article.

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
