# Peer review of "Probiotics and Achyranthes bidentata Polysaccharides Improve Growth Performance via Promoting Intestinal Nutrient Utilization and Enhancing Immune Function of Weaned Pigs"

_animals, 2021, doi:10.3390/ani11092617_

Round 1

Reviewer 1 Report

The experiment is good, but it is necessary wide review of spelling and writing errors. 

tanks a lot

Author Response

Reviewer#1

The experiment is good, but it is necessary wide review of spelling and writing errors. 

tanks a lot

Response: thank you for your comments. We have done it throughout the manuscript.

Reviewer 2 Report

Dear Authors,

The manuscript is well written and the experiment seems to be well conducted. Moreover, the Authors are working with a probiotic from they are on Lab associate with a traditional herb from their country. And thus, they hypothesized it could be alternative to in-feed antibiotics. Congrats!

Regarding the herbs and antibiotics the Authors may give more information about it in the M&M. At all, the Authors should work on the M&M to make it more informative to the reader.

  1. Introduction

Line 46-50: The authors should define antibiotics. Are they talking about growth promoter, prophylactic or therapeutic doses?

There is lot of misunderstanding in the literature about that.

  1. M&M

Line 69: Delete “healthy”

Line 70: instead use “about” include the standard deviation with the average weight.

Line 71: 75 mg/kg of the commercial product or the chemical actively compound?

What was the brand of the antibiotic?

Is this a growth promoter or a therapeutic dose?

The author should include all these information.

Line 80: The NRC recommendation is 1,50-1,35-1,23% SIDLys for first nursery phases diets…it seem the Authors did not used the just weaned recommendation. It is OK, however, it would be suitable to rephrase it.

Line 80-85: The Authors should include here information about the antibiotics. They may considerer a separate paragraph for the additives (ant, pro, abps)information.

Moreover, the herb is an oil or extract? It is classified as a phytogenic? It would be useful to add information about it to the reader.

Table 1: The Authors should present the Metabolizable Energy instead of digestible.

Table 1: The authors should include the SID Valine, Lactose, available phosphorus and total Calcium level in Calculated Nutrient level.

Table 1: The diet did not had a source of phosphorus neither of calcium? How did you reach the recommended level of Ca e P?

Line 109: Please in include the date. It is easier for the reader.

“At the conclusion of the trial (d XX), one pig from…”

Line 109: How did you choose the pig to collect blood? Based on BW or randomly…

Line 138: “Statistical analysis”, the Authors should detail how they did run these analyses. BW covariate? Pen or pig as experimental unit for the analysis? Etc.

  1. Results

Table 2: Suggestion, the Author should include the age of the pigs in title. …of piglets 23-XX day. Concerning performance it is always useful to know the age.

  1. Discussion

Line 207: There is no need for the abbreviation PWD.

Line 219: There is no need for the abbreviation TCM.

Line 266-268: The Authors should reference some papers that support the information about IL-2.

  1. Conclusion

OK.

Respectfully.

Author Response

Reviewer#2

The manuscript is well written and the experiment seems to be well conducted. Moreover, the Authors are working with a probiotic from they are on Lab associate with a traditional herb from their country. And thus, they hypothesized it could be alternative to in-feed antibiotics. Congrats!

Regarding the herbs and antibiotics the Authors may give more information about it in the M&M. At all, the Authors should work on the M&M to make it more informative to the reader.

  1. Introduction

Line 46-50: The authors should define antibiotics. Are they talking about growth promoter, prophylactic or therapeutic doses?

There is lot of misunderstanding in the literature about that.

Response: thank you for your advise. “in-feed antibiotics” is ususally considered as “antibiotic growth promoters (AGP)”, and we have added the information to reduce the misunderstanding in the revised manuscript.

  1. M&M

Line 69: Delete “healthy”

Response: thank you for your comments. We have done it.

Line 70: instead use “about” include the standard deviation with the average weight.

 Response: thank you for your comments. The standard deviation is for the weaning age but not for the average weight, so the “about” should be retained.

Line 71: 75 mg/kg of the commercial product or the chemical actively compound?

What was the brand of the antibiotic?

Is this a growth promoter or a therapeutic dose?

The author should include all these information.

 Response: thank you for your comments. Antibiotics used in our study is chlortetracycline hydrochloride belonging to AGP, which is provided by the Chia Tai Group and its available content is 15%.

Line 80: The NRC recommendation is 1,50-1,35-1,23% SIDLys for first nursery phases diets…it seem the Authors did not used the just weaned recommendation. It is OK, however, it would be suitable to rephrase it.

  Response: thank you for your comments. A good question. Low-protein diets has been popularized in animal production since 2019 in China and our country draw up corresponding standard for feed industry. Therefore, we added crystalline amino acids (Lys, Met, Thr and Trp) in our diet. As we all know, 1.23% of SIDLys in our formula just meet the NRC 2012 recommendation for 11-25 kg piglet, but this practice is commonly seen in the whole nursery period in our country. From weaned to nursery phase, only one diet were provided and the creep feed (1.50 or 1.35% of SID) just given during the lactation period after 7d of age. In conclusion, our diet formula not only met the requirement of NRC 2012 but also comply with the practice of Chinese swine production.

Line 80-85: The Authors should include here information about the antibiotics. They may considerer a separate paragraph for the additives (ant, pro, abps)information.

Moreover, the herb is an oil or extract? It is classified as a phytogenic? It would be useful to add information about it to the reader.

 Response: thank you for your comments. We have given supplementary information in the revised manuscript. Additionally, the herb used in our experiment is plant extract and has been commercialized in our country. The product is prepared by extraction and drying process and the its purity (99 %) was measured by HPLC.

Table 1: The Authors should present the Metabolizable Energy instead of digestible.

 Response: thank you for your comments. We have added the calculated ME in Table 1. As we all know, the ME and NE system is commonly used in Europe, DE or ME system in USA and DE system in China. So the calculated DE also retained in the table. Additionally, ME value for the ingredients (Glucose and Fatty powder) are not found in the standard, therefore we just provide the estimated ME value .

Table 1: The authors should include the SID Valine, Lactose, available phosphorus and total Calcium level in Calculated Nutrient level.

  Response: thank you for your comments. Low protein, amino acid-supplemented diet were prepared in our study. The CP is about 18% and we add Lys, Met, Thr and Trp in the diet, a lot of researches indicated that reducing CP and adding the four limited amino acids did not affect pig growth but reduced nitrogen emission and saved protien sources. In our study, we just consider four amino acids when reducing CP content, so we did not provided the calculated value for SID Valine. The calculated Lactose value can not be offerred due to lack of reference value in the standard. Indeed, we can not give the available phosphorus and total Calcium level in Calculated Nutrient level, because the ingredients for phosphorus and calcium are included in the commercial premix, and we can not obtain the calculated value due to the secrecy of premix formula.

Table 1: The diet did not had a source of phosphorus neither of calcium? How did you reach the recommended level of Ca e P?

 Indeed, we can not give the available phosphorus and total Calcium level in Calculated Nutrient level, because the ingredients for phosphorus and calcium are included in the commercial premix, and we can not obtain the calculated value due to the secrecy of premix formula.

Line 109: Please in include the date. It is easier for the reader.

“At the conclusion of the trial (d XX), one pig from…”

 Response: thank you for your comments. We have added it.

Line 109: How did you choose the pig to collect blood? Based on BW or randomly…

 Response: thank you for your comments. We have done it based on BW and we have added information in the revised manuscript.

Line 138: “Statistical analysis”, the Authors should detail how they did run these analyses. BW covariate? Pen or pig as experimental unit for the analysis? Etc.

 Response: thank you for your comments. We have added a detailed description in the revised manuscript.

  1. Results

Table 2: Suggestion, the Author should include the age of the pigs in title. …of piglets 23-XX day. Concerning performance it is always useful to know the age.

 Response: thank you for your comments. We have added the information in the revised manuscript.

  1. Discussion

Line 207: There is no need for the abbreviation PWD.

 Response: thank you for your advise. We have deleted it.

Line 219: There is no need for the abbreviation TCM.

 Response: thank you for your advise. We have deleted it.

Line 266-268: The Authors should reference some papers that support the information about IL-2.

  Response: thank you for your comments. We have added corresponding references in the revised manuscript.

 Conclusion

OK.

Respectfully.

Reviewer 3 Report

Suggestions

The manuscript address a very important issue and presents a valuable alternative to the use of antibiotics in pig production.

Simple summary.

L15: It is commonly seen that in-feed antibiotics is added to piglets diets because of …,

Please consider the following change: It is frequent to see that in-feed antibiotics are added to piglets' diets because of …,

L17:

L17 a "since July 1 in 2020 in China," Please remove in: since July 1 2020 in China

L17 b "Therefore, it urgent to develop some promising," please change to it is urgent to develop

Abstract.

L26: “120 weaned pigs,”

Could you please start the sentence using: One hundred and twenty weaned pigs; or you could say: Weaned pigs (120) were used ….

L26 “In our study, compared with the Con group” Please consider the following change: CON

Could you please indicate age and weight of weaned piglets in the abstract?

L32: “Interestingly, P-ABPS is more comparable to antibiotics than PRO or ABPS”.

Please consider the following change: Interestingly, P-ABPS effects were similar to those obtained with ANT than with PRO or ABPS.

L33-34: This sentence needs to be improved

“In conclusion, Dietary PRO or ABPS used alone or in combination (P-ABPS), the combined effect working better,”

Introduction.

L41-42: Please improve this sentence

“In order to increase the pigs weaned per sow per year (PSY) and improve the utilization efficiency of pigsty,”

L49a: “because of their side effects in humman health and environment “

Please consider the following change: because of their side effects in human health and the environment

L49b “Therefore, safe and efficient antibiotic alternatives for young pigs have become a global focus.”

Please consider the following change: Therefore, finding safe and efficient antibiotic alternatives for young pigs have become a global focus

Another alternative could be: Therefore, the use of safe and efficient antibiotic alternatives for young pigs have become a global focus

L54 “perfomence” [4-5]. please change to performance

L56: “Achyranthes bidentata (ABPS) possessed immuno-modulatory functions, and can be used as a diet …”

Please consider the following change Achyranthes bidentata (ABPS) possesses

L60: “…would improve piglets’ growth and its beneficial effect was better than their used alone,..”

Please consider the following change: would improve piglets’ growth and its beneficial effect would be better than when used alone,

L61: to in-feed anitibiotics please change to: in-feed antibiotics

Materials and Methods.

L70

“were randomly to 1 of 5 diets:”

I think the word assigned is missing. Please consider the following change were randomly assigned to 1 of 5 diets:

An alternative could be were randomized to 1 of 5 diets:

L132 “were measured” Please consider the following change: was measured

Statistical analysis

It is important to have a more detailed description of the statistical analysis.

Could you please indicate the experimental design?

Could you please indicate which is/are the experimental unit(s)?

Could you please describe the statistical models that were used?

Results

L148, L151, L152, L161, L164, L165, L176, L179, L180, L191, L194, L195, L201,

polysaccharide” Please change to polysaccharides

L185

“Compared with the Con and ANT groups, jejunal villus height (VH) in ABPS and P-ABPS were improved”

Please consider the following change: Compared with the CON and ANT groups, jejunal villus height (VH) in ABPS and P-184 ABPS was improved

L189 “In additional, sIgA in the jejunal mucosa of pigs fed ANT, ABPS and P-ABPS diets were greater”

Please consider the following change: Additionally, sIgA in the jejunal mucosa of pigs fed ANT, ABPS and P-ABPS diets was greater

Discussion

L207 Please change “sovle” to solve

L210 “P-ABPS supplementation showed the similar beneficial role on piglet growth performance 210 as ANT.”

Please consider the following change: P-ABPS supplementation showed a similar beneficial role on piglet growth performance 210 as ANT.

L225 Please change “expectaions” to expectations

L225-226 “was line with our expectaions and manifested that there existed an interaction of the combined use of the two products.” This asseveration could be misleading as apparently (it is not clear from section 2.7. Statistical analysis) you didn’t test interactions.

Please consider the following change: was in line with our expectations and suggest the existence of an interaction of the combined use of the two products.

or:

was lined/aligned with our expectations and suggest the existence of an interaction of the combined use of the two products.

L232 "products added promoted" gastrointestinal digestion I suggest to remove added

products promoted gastrointestinal digestion

L231-233 this paragraph is a little confusing needs revision

“… demonstrating that these products added promoted gastrointestinal digestion and absorption to enhance the nutrients utilization efficiency, which offered an evidence for improving piglet growth performance.”

Please consider the following change:

… demonstrating that these products promoted gastrointestinal digestion and absorption, enhancing the nutrients utilization efficiency, which is an evidence of the improvement in piglet growth performance.

L234 “were” please change to was

L235 “antiboitics” please change to antibiotics

L236 “intesinal” please change to intestinal

L234-236 I have the feeling that this paragraph needs a little improvement:

“The improvement of nutrient utilization were chiefly associated with reducing nutrient consumption of intestinal bacteria by in-feed antiboitics [25], producing organic acids and digestive enzymes by probiotics [17], and improving intesinal immune function by ABPS [6-7].”

Please consider the following change: The improvement of nutrient utilization was chiefly associated to reducing nutrient consumption of intestinal bacteria by in-feed antibiotics [25], producing organic acids and digestive enzymes by probiotics [17], and improving intestinal immune function by ABPS [6-7].

Author Response

Reviewer#3

The manuscript address a very important issue and presents a valuable alternative to the use of antibiotics in pig production.

Simple summary.

L15: It is commonly seen that in-feed antibiotics is added to piglets diets because of …,

Please consider the following change: It is frequent to see that in-feed antibiotics are added to piglets' diets because of …,

 Response: thank you for your comments. We have changed it.

L17:

L17 a "since July 1 in 2020 in China," Please remove in: since July 1 2020 in China

 Response: thank you for your comments. We have deleted it.

L17 b "Therefore, it urgent to develop some promising," please change to it is urgent to develop

 Response: thank you for your comments. We have added it.

Abstract.

L26: “120 weaned pigs,”

Could you please start the sentence using: One hundred and twenty weaned pigs; or you could say: Weaned pigs (120) were used ….

 Response: thank you for your comments. We have changed it.

L26 “In our study, compared with the Con group” Please consider the following change: CON

Could you please indicate age and weight of weaned piglets in the abstract?

 Response: thank you for your advise. We have changed it throughout the paper.

L32: “Interestingly, P-ABPS is more comparable to antibiotics than PRO or ABPS”.

Please consider the following change: Interestingly, P-ABPS effects were similar to those obtained with ANT than with PRO or ABPS.

Response: thank you for your advise. We accepted it.

L33-34: This sentence needs to be improved

“In conclusion, Dietary PRO or ABPS used alone or in combination (P-ABPS), the combined effect working better,”

Response: thank you for your comments. We have improved it.

Introduction.

L41-42: Please improve this sentence

“In order to increase the pigs weaned per sow per year (PSY) and improve the utilization efficiency of pigsty,”

Response: thank you for your comments. We have change this sentence to “Early weaning  can increase the pigs weaned per sow per year (PSY) and improve the utilization efficiency of pigsty, which has been commonly applied in the modern swine industry all over the world”

L49a: “because of their side effects in humman health and environment “

Please consider the following change: because of their side effects in human health and the environment

Response: thank you for your comments. We have changed them.

L49b “Therefore, safe and efficient antibiotic alternatives for young pigs have become a global focus.”

Please consider the following change: Therefore, finding safe and efficient antibiotic alternatives for young pigs have become a global focus

Another alternative could be: Therefore, the use of safe and efficient antibiotic alternatives for young pigs have become a global focus

Response: thank you for your advise. We have changed it.

L54 “perfomence” [4-5]. please change to performance

Response: thank you for your comments. We have corrected it.

L56: “Achyranthes bidentata (ABPS) possessed immuno-modulatory functions, and can be used as a diet …”

Please consider the following change Achyranthes bidentata (ABPS) possesses

Response: thank you for your comments. We accepted it.

L60: “…would improve piglets’ growth and its beneficial effect was better than their used alone,..”

Please consider the following change: would improve piglets’ growth and its beneficial effect would be better than when used alone,

Response: thank you for your comments. We have changed it.

L61: to in-feed anitibiotics please change to: in-feed antibiotics

Response: thank you for your comments. We have changed it.

Materials and Methods.

L70

“were randomly to 1 of 5 diets:”

I think the word assigned is missing. Please consider the following change were randomly assigned to 1 of 5 diets:

An alternative could be were randomized to 1 of 5 diets:

Response: thank you for your comments. We have changed it.

L132 “were measured” Please consider the following change: was measured

Response: thank you for your comments. We have changed it.

Statistical analysis

It is important to have a more detailed description of the statistical analysis.

Could you please indicate the experimental design?

Could you please indicate which is/are the experimental unit(s)?

Could you please describe the statistical models that were used?

 Response: thank you for your comments. We have added a detailed description in the revised manuscript.

Results

L148, L151, L152, L161, L164, L165, L176, L179, L180, L191, L194, L195, L201,

“polysaccharide” Please change to polysaccharides

L185

“Compared with the Con and ANT groups, jejunal villus height (VH) in ABPS and P-ABPS were improved”

Please consider the following change: Compared with the CON and ANT groups, jejunal villus height (VH) in ABPS and P-184 ABPS was improved

Response: thank you for your comments. We have changed it.

L189 “In additional, sIgA in the jejunal mucosa of pigs fed ANT, ABPS and P-ABPS diets were greater”

Please consider the following change: Additionally, sIgA in the jejunal mucosa of pigs fed ANT, ABPS and P-ABPS diets was greater

Response: thank you for your comments. We have changed it.

Discussion

L207 Please change “sovle” to solve

Response: thank you for your comments. We have changed it.

L210 “P-ABPS supplementation showed the similar beneficial role on piglet growth performance 210 as ANT.”

Please consider the following change: P-ABPS supplementation showed a similar beneficial role on piglet growth performance 210 as ANT.

Response: thank you for your comments. We have changed it.

L225 Please change “expectaions” to expectations

Response: thank you for your comments. We have changed it.

L225-226 “was line with our expectaions and manifested that there existed an interaction of the combined use of the two products.” This asseveration could be misleading as apparently (it is not clear from section 2.7. Statistical analysis) you didn’t test interactions.

Please consider the following change: was in line with our expectations and suggest the existence of an interaction of the combined use of the two products.

or:

was lined/aligned with our expectations and suggest the existence of an interaction of the combined use of the two products.

Response: thank you for your comments. We accept your advise and have changed it.

L232 "products added promoted" gastrointestinal digestion I suggest to remove added

products promoted gastrointestinal digestion

Response: thank you for your comments. We have changed it.

L231-233 this paragraph is a little confusing needs revision

“… demonstrating that these products added promoted gastrointestinal digestion and absorption to enhance the nutrients utilization efficiency, which offered an evidence for improving piglet growth performance.”

Please consider the following change:

… demonstrating that these products promoted gastrointestinal digestion and absorption, enhancing the nutrients utilization efficiency, which is an evidence of the improvement in piglet growth performance.

Response: thank you for your comments. We have changed it.

L234 “were” please change to was

Response: thank you for your comments. We have changed it.

L235 “antiboitics” please change to antibiotics

Response: thank you for your comments. We have changed it.

L236 “intesinal” please change to intestinal

Response: thank you for your comments. We have changed it.

L234-236 I have the feeling that this paragraph needs a little improvement:

“The improvement of nutrient utilization were chiefly associated with reducing nutrient consumption of intestinal bacteria by in-feed antiboitics [25], producing organic acids and digestive enzymes by probiotics [17], and improving intesinal immune function by ABPS [6-7].”

Please consider the following change: The improvement of nutrient utilization was chiefly associated to reducing nutrient consumption of intestinal bacteria by in-feed antibiotics [25], producing organic acids and digestive enzymes by probiotics [17], and improving intestinal immune function by ABPS [6-7].

Response: thank you for your comments. We have changed it.

Round 2

Reviewer 2 Report

Dear Authors,

In the Introduction: I still believe you should use “AGP” as you did in M&M. That would be more appropriated.

In the M&M: It is wild you made a formula without knowing the available Phosphorus. However, you could inform somehow that there was P and Ca in the premix.

You choose the pig “with the BW closest to the pen average”?

Respectfully.